# The Remediation in Enzyme’s Activities in Plants: Tea Waste as a Modifier to Improve the Efficiency of Growth of *Helianthus annuus* in Contaminated Soil

**DOI:** 10.3390/molecules27196362

**Published:** 2022-09-27

**Authors:** Sumeira Moin, Rafia Azmat, Waseem Ahmed, Abdul Qayyum, Hamed A. El-Serehy, Daniel Ingo Hefft

**Affiliations:** 1Department of Botany, Federal Urdu University of Arts, Sciences & Technology, Karachi 75300, Pakistan; 2Department of Chemistry, University of Karachi, Karachi 75270, Pakistan; 3Department of Horticulture, The University of Haripur, Haripur 22620, Pakistan; 4Department of Agronomy, The University of Haripur, Haripur 22620, Pakistan; 5Department of Zoology, College of Science, King Saud University, Riyadh l1451, Saudi Arabia; 6School of Chemical Engineering, University of Birmingham, Edgbaston Campus, Birmingham B15 2TT, UK

**Keywords:** Cd, tea waste, growth, enzymes activities, remediation

## Abstract

The remediation in plant enzymatic activities in Cd-contaminated soil was monitored through tea waste. Tea is an extensively used beverage worldwide with the release of a high quantity of tea waste utilized in the growing condition of *Helianthus annuus* on Cd metal contaminated soil. The study was a plan for the natural environmental condition in the greenhouse. For this purpose, four sets of plants were cultivated in triplicate and marked as (i) control, (ii) Cd stress plants, (iii) dry tea waste and Cd stress, and (iv) fresh tea waste and Cd stress. The improved efficiency of biochemical reactions in plants under Cd stress with tea waste treatment was the consequence of blocking Cd movement in the soil through adsorption on tea waste, showing that the tea waste effectively controls the mobility of Cd from the soil to the roots of the plants. Scan electron microscopy (SEM) validates the recovery of the leaves of the plants. The remediation of plant growth and enzyme activities such as amylase, peroxidase, nitrate reductase (NR), and nitrite reductase (NiR) under Cd metal-contaminated soil through tea waste was investigated. The source of tea waste in contaminated soil resulted in the recovery of the photosynthetic process and an improvement in amylase, NR, NiR, and peroxidase activities, thereby resulting in the recovery of pigments coupled with an increase in the biomass of the plants. It was suggested that tea waste acts as a good biosorbent of Cd and energy provider to the plants for normal enzyme activity under Cd stress and may be used by farmers in the future for safe and healthy crops as a cost-effective technology.

## 1. Introduction

Transformation in activities of molecules and enzymes displays the molecular mechanism operated for survival. The plant under abiotic stress related to its antioxidant defense system consists of various molecules like proline and enzymes, which play a significant role in the survival of plants to nullify the stress [1]. An enzyme is a particular protein: a chemical entity that acts as a catalyst to regulate the biological process rate at which the chemical reaction proceeds without being altered [2]. The role of enzymes in biochemical reactions within the cell is vital for cell life as they speed up the biochemical reactions of plants and aid an extensive array of significant plant functions. They accelerate the reaction at a very high rate under mild temperatures, pH, and pressure of the cells. The toxicity of heavy metals to plants is apparent, mainly linked to the blocking of the active site of the enzymes where the presence of the metal may displace some active ions necessary for the function of the cell [1,2,3].

Cd is a non-essential element for plant growth but a very active element in the environment [4] and is easily absorbed by the roots of the plant and transported to the shoot, uniformly distributed in different parts of the plants, maybe due to the false signal generated by this metal. The availability of Cd to plants is related to pH, soil organic matter, and redox potential [5]. Inhibition of processes of photosynthesis, enzyme activities, reduced plant pigments, and the generation of oxidative stress are the common symptoms of Cd toxicity to plant growth [6]. Chugh and Sawhney [7] observed the harmful effect of Cd on growth and amylase activity in the plant *Pisum sativum* cv. Bonneville at a very lower concentration of Cd metals (0.25 mM). Commonly, metal accumulation is indicated by the generation of reactive oxygen species or induction of enzymes in the defense mechanism by antioxidant enzymes such as peroxidase and catalase [8].

Gouia et al. [9] reported that the Cd also persuades a reduction in enzymes nitrate reductase after 24 h of exposure. The enzymes of nitrogen (N) metabolism are also affected by Cd toxicity, where a significant reduction in the activities of NR and NiR was observed, followed by a decline in the assimilation of nitrate in plant’s *Lycopersicon esculentum* [10]. Cd toxicity in plants also induced lipid peroxidation and oxidative stress, thereby increasing peroxidase activities [11]. The remediation in enzyme activities may be helpful in the ongoing metabolic process of the plant.

Tea waste is classified as a carbonaceous adsorbent that comprehends a recyclable energy substrate and nutrients, having special appearances such as a huge surface area, followed by fast kinetics of sorption that make it appropriate as a low-cost adsorbent for the removal of harmful inorganic and organic waste from wastewater. It is also a low-cost material for preparing active carbon [12]. Furthermore, tea waste as a non-toxic waste may be helpful in regulating enzyme activities under contaminated soil as it contains many significant ions, which is vital for growers primarily related to wastewater irrigation, by which soil becomes contaminated. The plant roots also absorb toxic metals, so the plant adopts survival strategies. The idea of using tea waste to restore agricultural land soil is very cost-effective and operative in controlling metal movement from soil to roots. The effect of tea waste is monitored as a remediation technology that can help to sustain enzymatic activities for the safe growth of plants. Therefore, this research was designed to investigate the metal obstructive from soil to roots through tea waste surface in the presence of Cd, where fresh and dry tea waste was applied in soil containing this metal while the impact was observed on the growth of the plants and enzymatic metabolism. The enzymes are essential and sensitive molecules of plant metabolism. Hence, four enzymes, including *amylase*, *peroxidase*, *nitrate reductase* (*NR*), and *nitrite reductase* (*NiR*), were monitored in non-treated (control) and treated plants (Cd and tea waste) as a critical part of plant metabolism.

## 2. Results and Discussion

### 2.1. Biomass Determination and Analysis of Photosynthesis

The transition metal Cd, non-essential for plants, was found to be increased due to anthropogenic activities, therefore available in the soil. The soil refers to contaminated soil that produces an inhibitory effect on the growing plants. The current investigation was planned to control the toxicity of the soil, thereby restricting the accumulation of Cd toxic metal into the plants. For this purpose, wet and dry tea waste was applied in Cd-contaminated soil, and the growth of the *Helianthus annuus* (*H. annuus*) was monitored. Cd toxicity in *H. annuus* appears to reduce the size of leaves, plant height, fresh and dry weight, and roots [13]. It was observed that at a shallow concentration of Cd like 1–5 ppm, the growth of the *H. annuus* was inhibited while it was recovered in Cd soil under tea waste application. It was observed that tea wastes act as an adsorbent to concentrate the metal on the surface, increase agricultural yields, and establish it as a cost-effective natural fertilizer [6,7,14]. It also increases soil fertility by supplying the nutrients it contains, followed by remediation of the heavy metal toxicity as the best engineering strategy for the plants to survive safely in contaminated soil. It improves the soil condition and makes the plants grow better than normal plants [15,16]. The decline in fresh and dry weights of the plants under Cd suggested less assimilation of C or decreased photosynthetic activity, which resulted from metal accumulation in roots. The addition of the tea waste recovers the plants’ photosynthetic system by checking the Cd metal movement under the soil to the roots of the plants (Figure 1, Figure 2, Figure 3 and Figure 4). The overall impact of tea waste appeared as an increase in the leaves area of plants, higher than the leaves of stress plants, followed by similar trends in shoot and root length (Figure 1 and Figure 2). Fresh and dry weight (FW and D.) of the *H. annuus* showed significant reduction under Cd metal stress, which may be associated with the toxic effect of the Cd on synthesis and carbon assembling due to the reduced leaf area of the plants (Figure 1 and Figure 3).

The soil remediation under tea waste application with 1–5 ppm of Cd results in the recovery of FW and DW. After harvesting the plants, the underground part of the plant was most seriously affected by Cd metal in contaminated soil. At the same time, the application of dry and wet Tea waste showed a positive effect, where a stable growth in plant d was observed in the comparison of control plants a and b, the Cd-treated plant. The delay in flowering was also recovered in tea waste plants (Figure 2).

### 2.2. Scan Electron Microscopy of the Helianthus annuus Plants

The results of Cd toxicity were also checked on the leaves and stomata of the *H. annuus* plants using Scan Electron microscopy, which showed that Cd alters the surface and stomata of the leaves, which were recovered under tea waste applications. It was observed in Figure 5 and Figure 6 Pb1 that the surface of leaves is altered with sunken stomata, while tea waste-treated plants showed improved structure. The stomata are a virtual channel in plants sensitive to environmental stress and play a key role in experiencing the processes of photosynthesis when CO_2_ is absorbed and oxygen evolves. The reduced size of the stomata at different concentrations of Cd is linked with reduced photosynthesis, coupled with the reduced synthesis of chloroplast pigments. The process of photosynthesis is imperative but more sensitive to plant development as it processes the CO_2_ into phytomass, directly linked with the stomata and plant biomass. The sunken stomata decreased photo- synthetic efficiency under Cd accumulation (Pb 1–5 ppm), while it improved stomatal conditions under treatment and improved the plant biomass and other metabolic pathways. It is evident from Figure 1 that when the plant was under stress, initially, it observed the water deficit related to the reduced leaf size and stomatal size followed by reduced transpiration rate. In this study, treatment with tea waste restored the stomata ultrastructure destroyed during Cd stress (Figure 5).

### 2.3. The Remediation of Enzymes as a Biomarker

In recent years, researchers have become very aware of the toxicity of heavy metals to plant metabolism from the primary to the secondary metabolic pathway [6,17]. Heavy metals are destructive substances for photosynthesis. They are involved in destabilizing enzymes, oxidizing photosystem II (PS II), and disrupting the electron transport chain and mineral metabolism [6]. Enzymes are the most sensitive and crucial part of the plant in biomass construction for the development of plants through photosynthesis. Any stress to plants primarily affects photosynthesis, thereby altering the metabolic pathway for their protective mechanisms. Analysis of four enzymes such as amylase, nitrate, nitrite reductase, and peroxidase showed an increase in peroxidase, while a decrease in amylase, nitrate, and nitrite reductase (Table 1, Table 2 and Table 3). The alteration in enzyme activity resulted in altered overall biochemical and biophysical reactions, including photosynthesis and enzyme activity [6]. The reduced activity of enzymes is related to heavy metal accumulation, where the immobilization of starch and nutrient sources becomes limited. The significant enzyme amylase, a crucial component for plant growth, showed lowered activity in Cd-treated plants.

In contrast, these enzymes play a vital role in converting the starch into disaccharides and sugar maltose, essential for providing energy to plants before starting photosynthesis. The remediation in amylase activity was observed when tea waste was applied to the pots containing Cd from 1 to 5 ppm, where amylase activity was also found to be increased (Table 1) from control plants, which indicates that tea waste acts as a natural biofertilizer to determine the movement of metal into roots and provide nutrients to the plants. Amylase is an enzyme of germination, present in germinating seeds and involved in the mobilization of starch reserves transported as sugars and utilized by the growing embryo. The reduced biomass and morphology are linked with the decrease in amylase activity under Cd stress (Table 1 and Figure 1, Figure 2 and Figure 3). The stimulation of amylase synthesis under tea waste treatment results in higher starch synthesis (Table 1 and Figure 6), which may be the results of the stimulation of gibberellic acid, as reported in the hypothesis of Boothby and Wright [18] that cytokinins in turn may overcome the inhibition and allow gibberellin to stimulate amylase synthesis in the presence of the inhibitor.

In plants, cell nitrogen assimilation is regulated by Nitrate reductase (NR), which works as a source of Nitric Oxide (NO) required for plant growth and resistance to abiotic and biotic stresses. The current study reflects the lower growth rate and low photosynthetic activity under Cd stress, which is also related to the decline in Nitrate (NR) and nitrite reductase (NiR) activities required to catalyze the reduction in nitrate to ammonium (Table 1, Table 2 and Table 3). Moreover, the regulation of NR and NiR gene expression by carbohydrates (C) and nitrogen (N) metabolites were reduced in Cd stress, due to which overall low biomass was observed (Table 3 and Figure 1, Figure 2, Figure 3, Figure 4 and Figure 6). It is also reported that NR-mediated NO generation also plays a crucial role in protecting plants from abiotic stresses through activating antioxidant enzymes. The exact impact of Cd was observed in the current study, where peroxidase increases to control oxidative stress and allow plants to detoxify these reactive oxygen species (ROS). The treatment results in the improvement in the activities of enzymes, due to which good growth was observed in contaminated soil.

Many researchers [19,20] have reported Cd toxicity on nitrate reductase enzymes in plants, as it is an important enzyme involved in producing nitric oxide required for good biomass accumulation. The nitrate and light stimulate enzyme synthesis, whereas part of the light is used in carbohydrate synthesis in photosynthesis. The reduced photosynthetic activity and low C assimilation are directly linked with the low activity of both enzymes (Table 1, Table 2 and Table 3) as a synthesis of NR encouraged by glucose and several other carbohydrate molecules (Figure 6). The results showed that the decline in nitrate reductase activity in Cd-treated plants was also linked with plant sugars that were also reduced due to the reduced chlorophyll pigments (Table 3 and Figure 4 and Figure 6), which was similar to the previous studies of Campbell [21] and Kaiser et al. [22].

The remediation of soil and plant metabolism by treatment with tea waste results in a good growth rate. NR activity expression and activity are controlled by light, temperature, pH, CO_2_, O_2_, water potential, and N source. The reduction in water uptake in the presence of heavy Cd is likely to result in a decrease in nitrate concentration in the xylem flux, which depends on the transpiration rate [23]. In this research, similar results were observed that the decreased activity of NR related to the reduced water contents, leaf structure, and closed stomata of the leaf. Peroxidase is the enzyme that responds to both types of stress to protect the plant. It involves scavenging the reactive oxygen species. Vicuna [24] reported the significance of peroxidase: it acts as a source of hydrogen peroxide (H_2_O_2_) coupled with the capability to scavenge it. He also discusses that high peroxidase activity increases the tolerance of plants to abiotic stress.

The increased activity of peroxidase in Cd-contaminated soil is commonly observed by many researchers [18,25], where an increase in activity nullifies the stress and protects the plant from the adverse effect of ROS, frequently generated as a result of Cd toxicity (Figure 4 and Figure 6; Table 2). It is usually involved in the biosynthesis of lignin to execute the plant stronger for survival under stress. The peroxidase activity is successfully restored in contaminated soil in the presence of tea waste, indicating that it acts as an excellent sorbent to adsorbed metal on its surface, thereby observing good plant growth.

### 2.4. Mechanism of Remediation in Enzymes

The wastewater contains several significant metals; some are substantial for plant growth while some are lethal to the plants such as Cd, As, Pb, which are non-essential to plants, have no biological function in plant metabolism and can even be poisonous in minute concentrations. The accumulation of Cd inhibits enzyme activity with the increase in peroxidase, as this enzyme belongs to the plant’s natural defense system. The current search results related to Cd metal toxicity are in accordance with earlier reports on Cd and other metals. A detailed literature search showed that reduced activity of NR and NiR may result in the direct interaction of the Cd and –SH group at the active site of both enzymes [26,27,28]. It is also reported that the decline in NR activity in bean, tomato, and peas encouraged by Cd metals is connected to the inhibition of NO_3_ uptake and translocation [29,30]. Metal toxicity (Cr) reduced nitrate reductase activity through impaired substrate utilization [31]. Mixing tea waste in soil contaminated with Cd showed the best remediating technique for monitoring metallic movement within soil through adsorption. It was suggested that the tea waste could bind an excess concentration of Cd metal in contaminated soil. Additionally, tea waste is a rich source containing many polyphenols and other nutrient elements that play a crucial role in soil fertility. The tea waste showed two prominent roles in agricultural soil (i) blocking the movement of the Cd ion through sorption on its surface within soil and (ii) providing additional nutrients to the soil. Both characteristics of the tea waste were observed in this investigation, where remediation in enzymes with extra nutrients also supports the growth of the plants over controlled plants. The remarkable growth of the plants under contaminated soil with normal metabolism suggested that this method could be considered a comparatively efficient with other remediation strategies and should be extensively used to remove organic and inorganic micropollutants from soil irrigated by wastewater.

## 3. Materials and Methods

### 3.1. Cultivation of the Plants and Experimental Design

The pots experiment was conducted after filling with soil, taken from the Nursery of the University of Karachi, and left for two days. The seeds of *H. annuus*, famous for their oil and nutritional values, were bought from the nursery of the University of Karachi. The seeds were sterilized with 70% ethanol and then soaked for 30 min in distilled water at room temperature. The experiments were carried out in a randomized design in five replicates of each treatment and kept in natural environments. The tea waste was collected from a canteen of the University of Karachi, and a part of that was dried. Plants were grown in four groups of pots. Each group contained five pots, and each pot was filled with 12 kg of air-dried soil. Only two groups, Pc (fresh tea waste) and Pd (dry tea waste), were filled with 12 kg of soil with 12 gm of fresh and dry tea waste. One set of the plants was filled with untreated soil, referred to as Pa (Control), whereas the other three sets (Pb, Pc, Pd) were treated with different concentrations of CdCl_2_, i.e., 1,2,3,4 and 5 ppm, respectively. Healthy seeds were sown. Plants were irrigated regularly. After 12 weeks of treatment with Cd^2+^, all plants were harvested. The biophysical and biochemical parameters of all sets of plants were analyzed.

### 3.2. Physical Measurements of the Soil and Plant

Initially, soil samples were subjected to Electrical conductivity (EC) and pH measurement before the cultivation of plants and after harvesting matured plants using conductivity and pH meter by dissolving soil in water, followed by filtration. The irrigation of plants was carried out as follows: (i) Pa: Control plants untreated with CdCl_2_, irrigated with tap water; (ii) Pb1–Pb5: plants treated with 1–5 ppm CdCl_2_ solution in 1 L of distilled water, 12 kg of soil followed by irrigation with the prepared solution; (iii) Pc1–Pc5: plants treated with 1–5 ppm CdCl_2_ solution in 1 L of distilled water to a pot filled with 12 kg of soil and 12 g of fresh tea waste, followed by irrigation with prepared solution; and (iv) Pd1–Pd5: plants treated with 1–5 ppm CdCl_2_ solution in 1 L of distilled, 12 kg of soil and 12 g of dry tea waste irrigated with the prepared solution. The physical measurements of all sets of plants were recorded after 12 weeks of treatments, including leaf area, fresh mass, dry masses, root/shoot length, moisture, and biochemical analyses via plant tissue samples in the laboratory.

### 3.3. Growth Measurement

During cultivation, the growth of all sets of plants was monitored regularly with irrigation up to 12 weeks, where climatic conditions were as follows (Temperature = 28–31 °C, Precipitation: 8–10%, Humidity: 45–52%, Wind: 27 km/h). The phytomass of all sets was determined as fresh weight and dry mass, followed by the simultaneous root/shoot length measurement using electrical balance and a regular scale, respectively. The electric oven was used to calculate the dry weight of leaf and root samples of all plants at 80 °C until a constant weight was achieved. Barrs and Weatherley [32] calculated the total water absorption by subtracting the dry weight from the fresh weight. The technique consists essentially of comparing the water content of leaves’ tissues as a fresh sample with the fully turgid water content and expressing the result on a percentage basis, thus:RT = {(FW − DW)/(TW − DW)} × 100 
where FW is initial fresh weight, TW is fresh turgid weight, DW is dry weight, and RT is the relative turgidity).

### 3.4. Determination of Pigments Content

Leaves pigments were determined by crushing 0.5 gm of the sample with acetone, as described by Maclachlan and Zalik [33] and centrifuging at 3000 rpm for 3 min until a colorless solution was obtained. The solution was then scanned for the spectrum of the pigments against acetone as a blank reagent. The pigment content was calculated using the formula in milligrams per gram of fresh weight.
$$Chlorophyll\;“a”\left(mg/gm\right)=\frac{\;12.3D_{663}-0.86D_{645}\;}{1000\times W}\times V$$
$$Chlorophyll\;“b”\frac{mg}{g}=\frac{19.3D_{645}-3.6D_{663}}{1000\times W}\times V$$
$$Carotenoid\frac{mg}{g}=\frac{7.6\;D_{480}-1.49\;D_{510}\;}{1000\times W}\times V$$
$$Xanthophyll\frac{mg}{g}=\frac{1.07\;D_{617}-0.094\;D_{430}}{1000\times W}\times V$$
where *D*_663_ = absorbance at 663 nm, *D*_645_ = absorbance at 645 nm, *V* = volume of acetone, *W* = Fresh weight of plant taken.

### 3.5. Analysis of Carbohydrate

Anthron reagent was used to analyze plant carbohydrate contents [34], where 0.5 g of a fresh sample of root and leaves were taken to prepare the extract by crushing it into 10 mL of Phosphate buffer. Then, the crushed sample was centrifuged for 15 min at 3000 rpm. Next, 10 mL of anthrone reagent was added to 1 mL of the supernatant or extracted in four different test tubes. Then, 4 mL of distilled water was added to each test tube and heated in a water bath for 15–20 min. All tubes were cooled until the complex’s green color appeared, whose absorbance was recorded by a (UV-1800A) Shimadzu spectrophotometer at 620 nm against a blank reagent.

### 3.6. Analysis of Reducing Sugar

The Miller [35] method was used to determine the reducing sugar in plants, where 0.5 g of plants sample crushed into 10 mL of Phosphate buffer, centrifuged for 15 min at 3000 rpm, and supernatant/extract of the sample was collected, followed by adding the dinitrosalicylic acid (DNS) reagent (2 mL) in each extract tube (1 mL), placed on a water bath until an orange color developed. Absorbance was recorded (UV-1800A) on a Shimadzu spectrophotometer at 540 nm. The concentration of the reducing sugars was measured with the help of the standard, prepared using standard glucose.

### 3.7. Estimation of Amylase Activity

Amylase activity was estimated by the method of Bernfeld [36]. One g of fresh leaves and roots were crushed into chilled ice, 10 mL of Phosphate buffer with pH 6.8, then centrifuged at 1000 rpm for 5 min. Then, 1 mL of 1% starch solution was added into 1 mL extract and incubated for 20 min at room temperature. After that, 2 mL of DNS was added and heated in a water bath till the color changed. After cooling, absorbance was recorded at 540 nm by a Shimadzu spectrophotometer.

### 3.8. Estimation of Peroxidase

Maehly and Chance [37] described the method to measure the activity by weighing 0.5 g of leaves and roots of fresh samples crushed with 5 mL of chilled ice phosphate buffer and then centrifuged for 10 min at 1600 rpm. The extract (0.1 mL) of each sample was collected in a test tube in which 2.1 mL of H_2_O, 0.16 mL of H_2_O_2_, 0.32 mL buffer and 0.32 mL of Pyrogallol solution were added; the tubes were chilled. The sample solution was incubated for 15 to 30 min at room temperature, and absorbance was noted at 420 nm, where the standard graph of peroxidase concentration was plotted against absorbance.

### 3.9. Determination of Nitrite Reductase

Ramarao et al. [38] reported the method for the determination of nitrite reductase activity in 0.5 g of fresh leaves and roots samples, which were crushed with 10 mL of 0.2 M Phosphate buffer (pH 7.5), and 0.02 M KNO_3_ added, then incubated in the dark at 32 °C for 1 h. In the test tube, add 0.5 mL of 1% Sulphanilamide in 3 N HCl, 1 mL of the incubated solution, and 0.02 N(1-Naphthyl)-ethylenediamine dihydrochloride, mixed thoroughly, then incubated at room temperature for 20 min. Absorbance was recorded at 542 nm on the (UV-1800A) Shimadzu spectrophotometer. The activity was represented as µmol NO_2_/h per g fresh weight.

### 3.10. Determination of Nitrate Reductase

The activity of the enzyme Nitrate reductase was estimated by the method used in [39]. Then, 0.5 g of fresh leaves and roots samples were crushed into 10 mL of 0.2 M Phosphate buffer (pH 5) and 1 mL of 0.5 mM Sodium nitrite and placed in desiccators for 5 min. Aluminum foil was used to wrap the tubes before incubating at 30 °C on the water bath and shaking slowly for 30 min till boiling. In 1 mL of incubated solution, 1 mL of 1% sulfanilamide in 1N HCl and 1 mL of 0.02 N (1-Naphthyl)-ethylenediamine dihydrochloride were added. The sample was incubated at room temperature for 20 min. Absorbance was recorded at 540 nm on a (UV-1800A) Shimadzu spectrophotometer against the standard concentration. The activity was represented as µmol NO_2_/h per g fresh weight.

### 3.11. Scanning Electron Microscopy of Surface of Leaves

Scanning electron microscopy (SEM) of leaves was conducted to ensure the remediation of processes of photosynthesis through the surface of leaves and stomata, as described by Pathan et al. [40]. SEM mode is secondary electrons emitted by atoms excited via the electron beam. This method focuses an electron beam on the upper surface, creating an image. The electron beam interacts on the sample surface, generating several signals which can be used to acquire evidence related to the surface topography and composition.

### 3.12. Statistical Analysis

The data obtained were subjected to statistical analysis by using IBM. 20 version of SPSS. This study used one-way ANOVA, student *t*-test, and LSD. test *± Standard deviation of three replicates, Asterisks (*) represent significant differences (*p* < 0.05); double asterisks (**) represent highly significant differences (*p* < 0.01).

## 4. Conclusions

The current research suggests that enzymes are the most sensitive and essential part of plant metabolism for plant growth, where normal growth is linked with regular enzyme activity. It was concluded that the restoration of the growth of plants and enzyme activities under Cd contamination through tea waste suggested that metal-contaminated soil may also act as adequate fertile soil, which the farmer can easily adopt to increase the yield and save their crops. It was also concluded that tea waste acts (dry or fresh) as an excellent organic biosorbent to monitor the movement of non-essential metal ions from soil to root for the effective cultivation of crops. Tea waste has been proved as an easy, cost-effective, and suitable for soil remediation, leading to remediation in a whole metabolic path through enzyme recovery. Moreover, this process integrates waste management of toxic inorganic elements and organic waste in a beneficial way, where the danger associated with both wastes is removed. It may be used by farmers in the future for safe and healthy crops as a cost-effective technology.

## Figures and Tables

**Figure 1 molecules-27-06362-f001:**
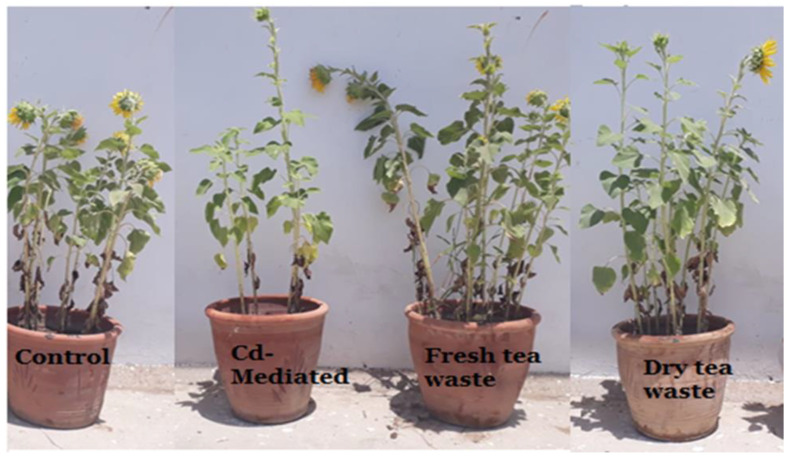
Morphological changes in shoot length of control, Cd stress, fresh and dry tea waste treated of *H. annuus* plants.

**Figure 2 molecules-27-06362-f002:**
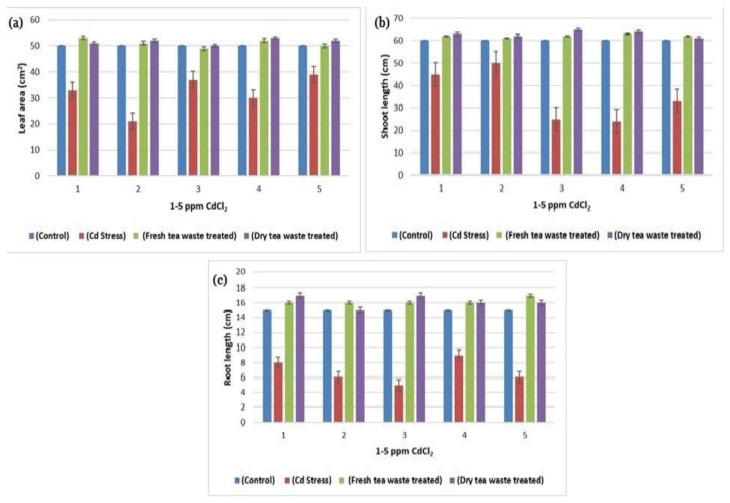
(**a**) A plot of comparison between Leaf area of control, Cd stress, fresh and dry tea waste treated of *H. annuus* plants (**b**) A plot of comparison between Shoot length of control, Cd stress, fresh and dry tea waste treated of *H. annuus* plants (**c**) A plot of comparison between Root length of control, Cd stress, fresh and dry tea waste treated of *H. annuus* plants.

**Figure 3 molecules-27-06362-f003:**
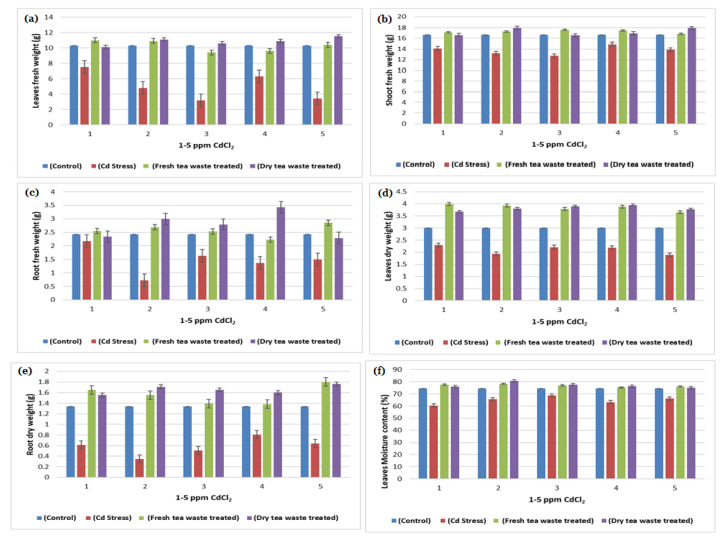
(**a**) A plot of comparison between Leaves fresh weight of control, Cd stress, fresh and dry tea waste treated of *H. annuus* plants (**b**) A plot of comparison between Shoot fresh weight of control, Cd stress, fresh and dry tea waste treated of *H. annuus* plants (**c**) A plot of comparison between Root fresh weight of control, Cd stress, fresh and dry tea waste treated of *H. annuus* plants (**d**) A plot of comparison between Leaves dry weight of control, Cd stress, fresh and dry tea waste treated of *H. annuus* plants (**e**) A plot of comparison between Shoot dry weight of control, Cd stress, fresh and dry tea waste treated of *H. annuus* plants (**f**) A plot of comparison between Root dry weight of control, Cd stress, fresh and dry tea waste treated of *H. annuus* plants.

**Figure 4 molecules-27-06362-f004:**
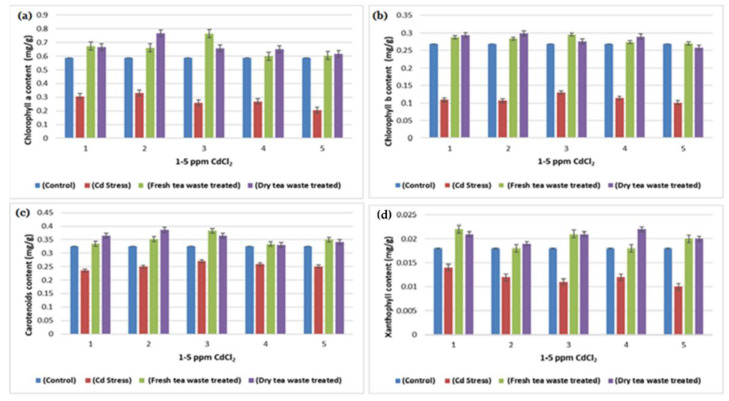
(**a**) A plot of comparison between Chlorophyll (**a**) content of control, Cd stress, fresh and dry tea waste treated of *Helianthus annuus* plants (**b**) A plot of comparison between Chlorophyll (**b**) content of control, Cd stress, fresh and dry tea waste treated of *H. annuus* plants (**c**) A plot of comparison between Carotenoids content of control, Cd stress, fresh and dry tea waste treated of *H. annuus* plants (**d**) A plot of comparison between Xanthophyll content of control, Cd stress, fresh and dry tea waste treated of *H. annuus* plants.

**Figure 5 molecules-27-06362-f005:**
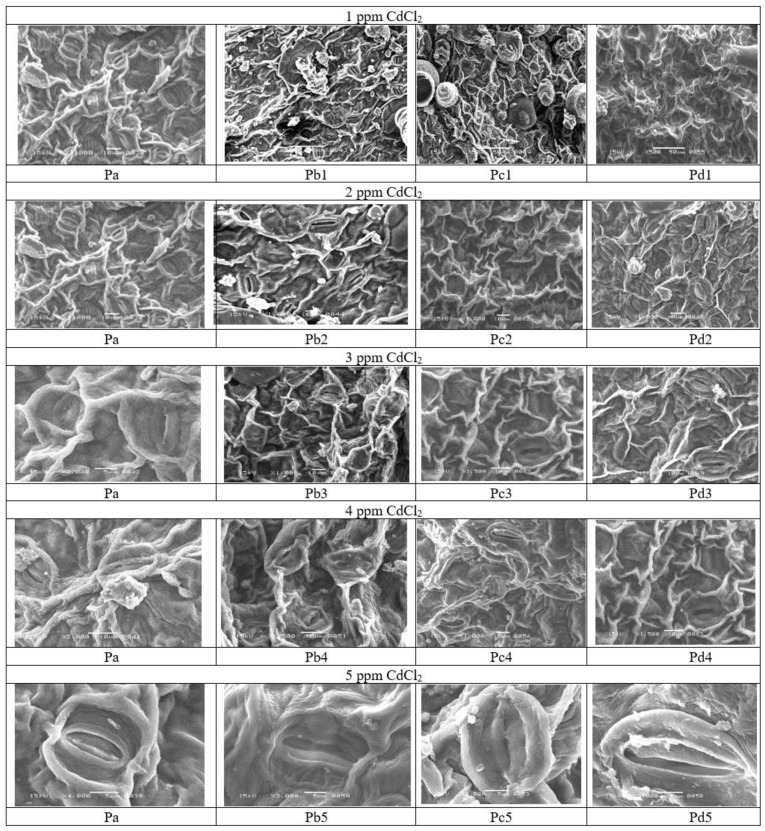
Scan Electron microscopy of leaves and stomata of *H. annuus* which showed the recovery of leaves in presence of tea waste where (Pa) showed the structure of leaves of control plants, (Pb) showed the structure of leaves under Different percentage of Cd, (Pc) showed the structure of leaves under wet Tea waste while (Pd) showed the structure of leaves stomata under dry Tea waste.

**Figure 6 molecules-27-06362-f006:**
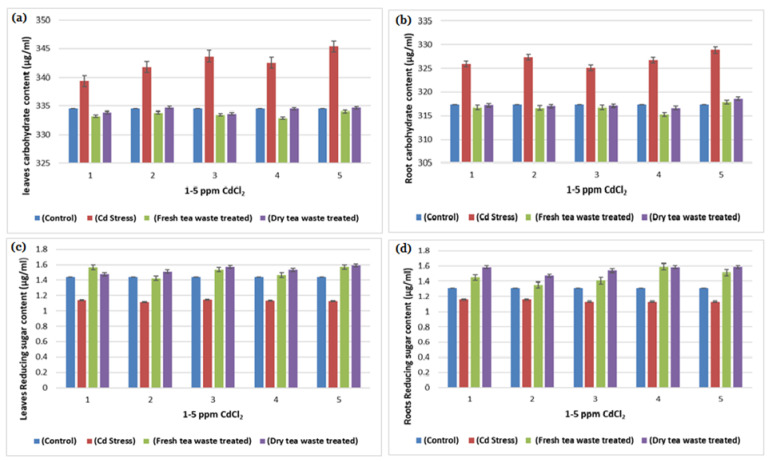
(**a**) A plot of comparison between Carbohydrate content of Leaves of control, Cd stress, fresh and dry tea waste treated of *H. annuus* plants (**b**) A plot of comparison between Roots Carbohydrate content of control, Cd stress, fresh and dry tea waste treated of *H. annuus* plants (**c**) A plot of comparison between Reducing sugar content of Leaves of control, Cd stress, fresh and dry tea waste treated of *H. annuus* plants (**d**) A plot of comparison between Reducing sugar content of control, Cd stress, fresh and dry tea waste treated of *H. annuus* plants.

**Table 1 molecules-27-06362-t001:** A comparative study of Amylase activity of *Helianthus annuus* Pa, Pb, Pc and Pd showing the impact of fresh and dry tea waste on plant Pc and Pd.

Plants Sample	Amylase Activity Content (µg/mL)	
	Leaves	Roots
**1 ppm CdCl_2_**		
Pa (Control)	37.46 ± 1.2 **	41.06 ± 0.1 **
Pb1 (Cd Stress)	21.80 ± 0.7 **	12.07 ± 0.8 **
Pc1 (Fresh tea waste treated)	38.32 ± 1.3 **	45.53 ± 0.4 **
Pd1 (Dry tea waste treated)	39.98 ± 0.4 **	45.33 ± 0.0 **
**2 ppm CdCl_2_**		
Pa (Control)	37.46 ± 0.1 **	41.06 ± 0.0 **
Pb2 (Cd Stress)	23.53 ± 0.2 **	13.26 ± 2.1 **
Pc2 (Fresh tea waste treated)	40.06 ± 0.8 **	39.86 ± 2.2 **
Pd12 (Dry tea waste treated)	41.65 ± 0.0 **	48.20 ± 0.8 **
**3 ppm CdCl_2_**		
Pa (Control)	37.46 ± 0.3 **	41.06 ± 1.1 **
Pb3 (Cd Stress)	24.07 ± 3.1 **	18.73 ± 1.3 **
Pc3 (Fresh tea waste treated)	37.21 ± 2.0 **	37.86 ± 0.5 **
Pd3 (Dry tea waste treated)	38.67 ± 0.1 **	47.40 ± 0.2 **
**4 ppm CdCl_2_**		
Pa (Control)	37.46 ± 0.2 **	41.06 ± 1.9 **
Pb4 (Cd Stress)	21.67 ± 0.0 **	19.80 ± 0.4 **
Pc4 (Fresh tea waste treated)	41.67 ± 1.7 **	44.53 ± 0.9 **
Pd4 (Dry tea waste treated)	42.06 ± 0.0 **	44.67 ± 0.4 **
**5 ppm CdCl_2_**		
Pa (Control)	37.46 ± 0.1 **	41.06 ± 0.9 **
Pb5 (Cd Stress)	25.20 ± 0.8 **	13.47 ± 0.1 **
Pc5 (Fresh tea waste treated)	43.13 ± 1.7 **	46.88 ± 0.5 **
Pd5 (Dry tea waste treated)	44.47 ± 0.3 **	46.47 ± 1.6 **

± Standard deviation of three replicates, the significance difference is less than *p* = 0.05, ** *p* < 0.01.

**Table 2 molecules-27-06362-t002:** A comparative study of Peroxidase activity content of *H. annuus* Pa, Pb, Pc and Pd showing the impact of fresh and dry tea waste on plant Pc and Pd.

Plants Sample	Peroxidase Activity Content (µg/mL)	
	Leaves	Roots
**1 ppm CdCl_2_**		
Pa (Control)	83.53 ± 0.7 **	61.53 ± 0.1 **
Pb1 (Cd Stress)	92.69 ± 0.5 **	72.61 ± 0.7 **
Pc1 (Fresh tea waste treated)	78.46 ± 0.1 **	60.92 ± 0.6 **
Pd1 (Dry tea waste treated)	74.76 ± 1.5 **	61.46 ± 0.4 **
**2 ppm CdCl_2_**		
Pa (Control)	83.53 ± 0.2 **	61.53 ± 0.1 **
Pb2 (Cd Stress)	90.15 ± 0.7 **	70.92 ± 0.5 **
Pc2 (Fresh tea waste treated)	80.46 ± 0.0 **	60.15 ± 1.2 **
Pd2 (Dry tea waste treated)	81.69 ± 1.7 **	61.23 ± 0.0 **
**3 ppm CdCl_2_**		
Pa (Control)	83.53 ± 0.2 **	61.53 ± 0.2 **
Pb3 (Cd Stress)	93.23 ± 2.1 **	72.53 ± 0.0 **
Pc3 (Fresh tea waste treated)	82.46 ± 0.3 **	62.92 ± 1.2 **
Pd3 (Dry tea waste treated)	82.76 ± 0.1 **	61.61 ± 0.4 **
**4 ppm CdCl_2_**		
Pa (Control)	83.53 ± 2.7 **	61.53 ± 0.4 **
Pb4 (Cd Stress)	95.69 ± 0.9 **	74.69 ± 0.2 **
Pc4 (Fresh tea waste treated)	82.30 ± 0.2 **	61.23 ± 0.6 **
Pd4 (Dry tea waste treated)	83.46 ± 1.7 **	61.38 ± 0.0 **
**5 ppm CdCl_2_**		
Pa (Control)	83.53 ± 0.0 **	61.53 ± 0.3 **
Pb5 (Cd Stress)	89.92 ± 0.3 **	74.92 ± 0.8 **
Pc5 (Fresh tea waste treated)	82.92 ± 1.4 **	61.46 ± 0.1 **
Pd5 (Dry tea waste treated)	83.69 ± 2.7 **	62.15 ± 1.3 **

± Standard deviation of three replicates, the significance difference is less than *p* = 0.05, ** *p* < 0.01.

**Table 3 molecules-27-06362-t003:** A comparative study of Nitrite reductase activity and Nitrate reductase activity contents of *H. annuus* Pa, Pb, Pc and Pd showing impact of fresh and dry tea waste on plant Pc and Pd.

Plants Sample	Nitrite Reductase Activity Content (µg/mL)		Nitrate Reductase Activity Content (µg/mL)	
	Leaves	Roots	Leaves	Roots
**1 ppm CdCl_2_**				
Pa (Control)	402.8 ± 0.9 **	263.4 ± 0.1 **	45.6 ± 0.0 **	38.3 ± 0.3 **
Pb (Cd Stress)	310.2 ± 1.6 **	186.8 ± 0.3 **	28.3 ± 0.1 **	22.6 ± 0.1 **
Pc (Fresh tea waste treated)	404.2 ± 0.1 **	265.6 ± 1.7 **	46.7 ± 0.3 **	39.3 ± 0.5 **
Pd (Dry tea waste treated)	403.2 ± 0.3 **	264.6 ± 0.1 **	45.9 ± 0.1 **	38.6 ± 1.5 **
**2 ppm CdCl_2_**				
Pa (Control)	402.8 ± 0.2 **	263.4 ± 0.0 **	45.6 ± 1.5 **	38.3 ± 0.6 **
Pb2 (Cd Stress)	308.8 ± 1.4 **	184.2 ± 0.1 **	24.1 ± 0.1 **	22.6 ± 0.2 **
Pc2 (Fresh tea waste treated)	402.9 ± 0.4 **	264.4 ± 0.6 **	45.9 ± 0.8 **	41.5 ± 1.7 **
Pd2 (Dry tea waste treated)	403.5 ± 0.6 **	265.7 ± 0.2 **	46.8 ± 0.2 **	42.9 ± 0.5 **
**3 ppm CdCl_2_**				
Pa (Control)	402.8 ± 0.2 **	263.45 ± 0.1 **	45.6 ± 1.6 **	38.3 ± 0.8 **
Pb3 (Cd Stress)	305.4 ± 0.1 **	181.36 ± 0.3 **	25.4 ± 1.1 **	23.8 ± 0.6 **
Pc3 (Fresh tea waste treated)	403.6 ± 0.4 **	263.81 ± 1.7 **	44.9 ± 0.2 **	39.6 ± 1.3 **
Pd3 (Dry tea waste treated)	404.2 ± 0.8 **	265.63 ± 0.3 **	46.2 ± 0.7 **	40.6 ± 0.3 **
**4 ppm CdCl_2_**				
Pa (Control)	402.8 ± 1.2 **	263.4 ± 0.2 **	45.6 ± 0.1 **	38.3 ± 0.1 **
Pb4 (Cd Stress)	307.4 ± 0.4 **	185.2 ± 0.1 **	29.3 ± 0.9 **	25.6 ± 0.9 **
Pc4 (Fresh tea waste treated)	404.2 ± 0.9 **	262.3 ± 1.2 **	45.5 ± 0.2 **	42.9 ± 0.0 **
Pd4 (Dry tea waste treated)	404.5 ± 1.5 **	264.0 ± 0.8 **	47.1 ± 1.3 **	43.1 ± 0.1 **
**5 ppm CdCl_2_**				
Pa (Control)	402.8 ± 0.3 **	263.4 ± 0.4 **	45.6 ± 0.4 **	38.3 ± 0.1 **
Pb (Cd Stress)	308.4 ± 2.1 **	183.8 ± 1.8 **	24.1 ± 0.2 **	23.1 ± 0.5 **
Pc5 (Fresh tea waste treated)	404.6 ± 1.6 **	264.8 ± 0.1 **	46.5 ± 1.7 **	39.9 ± 1.4 **
Pd5 (Dry tea waste treated)	405.7 ± 0.2 **	264.09 ± 1.2 **	46.9 ± 0.3 **	40.1 ± 1.5 **

± Standard deviation of three replicates, the significance difference is less than *p* = 0.05, ** *p* < 0.01.

## Data Availability

All Data of the current based in finding of research projected conducted.

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
