# Peer review of "The Remediation in Enzyme’s Activities in Plants: Tea Waste as a Modifier to Improve the Efficiency of Growth of *Helianthus annuus* in Contaminated Soil"

_molecules, 2022, doi:10.3390/molecules27196362_

Round 1
Reviewer 1 Report
1. In page 1 line 39-40, kindly insert reference(s)
2. Kindly delete the sentence, “the plant under abiotic stress related ………. _ survival to reverse the stress” (in line 40-42) as it is a repetition of second sentence in the introductory part.
3. In page 2 line 54, kindly delete 0.25mM
4. In page 2 line 57, kindly use correct reference citation format for the journal “Gouia et al. (2000)”. More so, the refence cited above is missing in the list of main refence.
5. In page 2 line 58-60, Kindly write the full meaning of “N” and leave only the abbreviation on N.R., NiR, as you have stated their meaning in your abstract
6. Kindly recast the sentences in line 69-76
7. Kindly state the percentage of the ethanol used in line 82 page 2
8. In line 90 page 2, it should be 1, 2, 3, 4 and 5 ppm not 1,2.3,4,5 ppm
9. In line 91 page 2, the sentence “After 12 weeks of treatment of Cd” it should be “After 12 weeks of treatment with Cd
10. In line 92 page2, kindly replace the first “and” with “the”
11. In line 92 page2, kindly delete in the following way
12. In section 2.1, kindly provide references to back up your method.
13. In line 98-103, kindly provide references
14. In line 109, kindly replace “was” with “were”
15. In line 110, …… were as follows (temperature …..
16. In line 115-116, the sentence should be” Barres & Weatherly [11] method was used to calculate the total water absorption
17. In line 148, it should be “Miller [14] method was used to determine the reducing sugar in plants ……… you don’t have to state everything in details please
18. In section 2.7, you avoid stating sentence with number. Kindly correct it place/
19. In line 161, kindly delete “(UV-1800A) Shimadzu spectrophotometer”. In addition, delete the last sentence as well.
20. In line 275, delete “and increasing antioxidant”
21. In line 281, fig. should be written in full
22. In line 288-292, insert references as the sentences are strong statements that requires citations.
23. In line 292-293, kindly recast the sentence
24. In line 302-311, please references please as the statements are strong and required citation
25. Section 3.4 “Mechanism of remediation….” I thought the authors are presenting their results findings first before discussion, as this section is completely discussion. Kindly move it under discussion.
26. In line 314, delete therefore”
27. Kindly rewrite the conclusion in order to reflect your result findings. You are to conclude based on your results.
Author Response
- In page 1 line 39-40, kindly insert reference(s)
Resolved: Thanks, corrected as suggested
- Kindly delete the sentence, “the plant under abiotic stress related ………. _ survival to reverse the stress” (in line 40-42) as it is a repetition of second sentence in the introductory part.
Resolved: Thanks, corrected as suggested
- In page 2 line 54, kindly delete 0.25mM
Resolved: Thanks, corrected as suggested
- In page 2 line 57, kindly use correct reference citation format for the journal “Gouia et al. (2000)”. More so, the refence cited above is missing in the list of main refence.
Resolved: Thanks, reference added
- In page 2 line 58-60, Kindly write the full meaning of “N”
Resolved: Thanks, corrected as suggested
and leave only the abbreviation on N.R., NiR, as you have stated their meaning in your abstract
Resolved: Thanks, corrected as suggested
- Kindly recast the sentences in line 69-76
Resolved: recast it, as suggested
- Kindly state the percentage of the ethanol used in line 82 page 2
Resolved: Thanks, corrected as suggested
- In line 90 page 2, it should be 1, 2, 3, 4 and 5 ppm not 1,2.3,4,5 ppm
Resolved: Thanks, corrected as suggested
- In line 91 page 2, the sentence “After 12 weeks of treatment of Cd” it should be “After 12 weeks of treatment with Cd
Resolved: Thanks, corrected as suggested
- In line 92 page2, kindly replace the first “and” with “the”
Resolved: Thanks, corrected as suggested
- In line 92 page2, kindly delete in the following way
Resolved: Thanks, corrected as suggested
- In section 2.1, kindly provide references to back up your method.
Resolved: references are added
- In line 98-103, kindly provide references
Resolved: references are added
- In line 109, kindly replace “was” with “were”
Resolved: Thanks, corrected as suggested
- In line 110, …… were as follows (temperature …..
Resolved: Thanks, corrected as suggested
- In line 115-116, the sentence should be” Barres & Weatherly [11] method was used to calculate the total water absorption
Resolved: Thanks, corrected as suggested
- In line 148, it should be “Miller [14] method was used to determine the reducing sugar in plants ……… you don’t have to state everything in details please
Resolved: Thanks, corrected as suggested
- In section 2.7, you avoid stating sentence with number. Kindly correct it place/
Resolved: Thanks, corrected as suggested
- In line 161, kindly delete “(UV-1800A) Shimadzu spectrophotometer”. In addition, delete the last sentence as well.
Resolved: Thanks, corrected as suggested
- In line 275, delete “and increasing antioxidant”
Resolved: Thanks, deleted as suggested
- In line 281, fig. should be written in full
Resolved: Thanks, corrected as suggested
- In line 288-292, insert references as the sentences are strong statements that requires citations.
Resolved: Thanks, added as suggested
- In line 292-293, kindly recast the sentence
Resolved: sentence is recast
- In line 302-311, please references please as the statements are strong and required citation
Resolved: Thanks, added as suggested
- Section 3.4 “Mechanism of remediation….” I thought the authors are presenting their results findings first before discussion, as this section is completely discussion. Kindly move it under discussion.
Resolved: Thanks, This is the part of Discussion in results discussion section
- In line 314, delete therefore”
Resolved: Corrected as suggested
- Kindly rewrite the conclusion in order to reflect your result findings. You are to conclude based on your results.
Resolved: the whole conclusion is revised as per suggestion
Reviewer 2 Report
The manuscript has lots of good quality work and the authors carried out massive experiments to investigate the remediation of plants' growth and enzymes activities like amylase, peroxidase, nitrate reductase, and nitrite reductase under Cd metal contaminated soil through tea waste. However, some clarifications in the paper and improvement in English presentation are suggested.
Introduction: What are the properties of tea waste? This needs to be highlighted in the introductory paragraph.
L39: Not clear “…significant functions in the animal and body”
L41: biotic and abiotic stresses
L51: oxidative stresses are
L57, 156, 183: Please follow the uniform citation pattern.
L59: Better to abbreviate “NR”
L82: absolute ethanaol?
L126: spectrum scan? Please specify
L224: Figure 2-4 & 6. What kind of error bars are represented?? standard deviation / standard error or else.
L287-311: Better to move this paragraph to the discussion section.
L341: Adjust Figure 5 on a single page.
L393: Check the Table number.
Author Response
The manuscript has lots of good quality work and the authors carried out massive experiments to investigate the remediation of plants' growth and enzymes activities like amylase, peroxidase, nitrate reductase, and nitrite reductase under Cd metal contaminated soil through tea waste.
However, some clarifications in the paper and improvement in English presentation are suggested.
Resolved: Overall English corrections is done
Introduction: What are the properties of tea waste? This needs to be highlighted in the introductory paragraph.
Resolved: Added as suggested with reference
L39: Not clear “…significant functions in the animal and body”
Resolved: corrected as suggested
L41: biotic and abiotic stresses
Resolved: corrected as suggested
L51: oxidative stresses are
Resolved: corrected as suggested
L57, 156, 183: Please follow the uniform citation pattern.
Resolved: corrected as suggested
L59: Better to abbreviate “NR”
Resolved: corrected as suggested
L82: absolute ethanaol?
Resolved: corrected as suggested
L126: spectrum scan? Please specify
Resolved: corrected as suggested
L224: Figure 2-4 & 6. What kind of error bars are represented?? standard deviation / standard error or else.
Resolved: Standard deviation
L287-311: Better to move this paragraph to the discussion section.
Resolved: This is already in last as a discussion part
L341: Adjust Figure 5 on a single page.
Resolved: corrected as suggested
L393: Check the Table number.
Resolved: corrected as suggested